# The Unfolded Protein Response Is Associated with Cancer Proliferation and Worse Survival in Hepatocellular Carcinoma

**DOI:** 10.3390/cancers13174443

**Published:** 2021-09-03

**Authors:** Ankit Patel, Masanori Oshi, Li Yan, Ryusei Matsuyama, Itaru Endo, Kazuaki Takabe

**Affiliations:** 1Department of Surgical Oncology, Roswell Park Comprehensive Cancer Center, Buffalo, NY 14263, USA; ankit.patel@roswellpark.org (A.P.); masa1101oshi@gmail.com (M.O.); 2Department of Gastroenterological Surgery, Graduate School of Medicine, Yokohama City University, Yokohama 236-0004, Japan; ryusei@yokohama-cu.ac.jp (R.M.); endoit@yokohama-cu.ac.jp (I.E.); 3Department of Biostatistics & Bioinformatics, Roswell Park Comprehensive Cancer Center, Buffalo, NY 14263, USA; li.yan@roswellpark.org; 4Department of Surgery, Jacobs School of Medicine and Biomedical Sciences, State University of New York, Buffalo, NY 14263, USA; 5Department of Breast Surgery and Oncology, Tokyo Medical University, Tokyo 160-8402, Japan; 6Department of Digestive and General Surgery, Graduate School of Medicine and Dental Sciences, Niigata University, Niigata 951-8520, Japan; 7Department of Breast Surgery, School of Medicine, Fukushima Medical University, Fukushima 960-1295, Japan

**Keywords:** unfolded protein, hepatocellular cancer, GSVA, unfolded protein score

## Abstract

**Simple Summary:**

We studied the association between the unfolded protein response (UPR) and carcinogenesis, cancer progression, and survival in hepatocellular carcinoma (HCC). We studied 655 HCC patients from 4 independent cohorts using an UPR score. The UPR was enhanced as normal liver became cancerous and as HCC advanced in stage. The UPR was correlated with cancer cell proliferation that was confirmed by multiple parameters. Significantly, a high UPR score was associated with worse patient survival. Interestingly, though UPR was associated with a high mutational load, it was not associated with immune response, immune cell infiltration, or angiogenesis. To our knowledge, this is the first study to investigate the clinical relevance of the unfolded protein response in HCC.

**Abstract:**

Hepatocellular carcinoma is a leading cause of cancer death worldwide. The unfolded protein response (UPR) has been revealed to confer tumorigenic capacity in cancer cells. We hypothesized that a quantifiable score representative of the UPR could be used as a biomarker for cancer progression in HCC. In this study, a total of 655 HCC patients from 4 independent HCC cohorts were studied to examine the relationships between enhancement of the UPR and cancer biology and patient survival in HCC utilizing an UPR score. The UPR correlated with carcinogenic sequence and progression of HCC consistently in two cohorts. Enhanced UPR was associated with the clinical parameters of HCC progression, such as cancer stage and multiple parameters of cell proliferation, including histological grade, mKI67 gene expression, and enrichment of cell proliferation-related gene sets. The UPR was significantly associated with increased mutational load, but not with immune cell infiltration or angiogeneis across independent cohorts. The UPR was consistently associated with worse survival across independent cohorts of HCC. In conclusion, the UPR score may be useful as a biomarker to predict prognosis and to better understand HCC.

## 1. Introduction

Primary liver cancer is the sixth most common cancer worldwide with hepatocellular carcinoma (HCC) compromising the majority of cases [1]. Incidence and mortality rates have decreased in high-risk regions in the worlds, yet prognosis remains poor, with an expected 5-year survival rate less than 40% [2]. Improved outcomes may be achieved in the 10–15% of patients in whom surgical resection is possible, but the majority of patients with a nonresectable disease have limited benefit from systemic chemotherapy [3]. A biomarker based on tumor biology can help optimize treatment choices when linked to prognosis.

Cancer cells have the unique ability to evoke adaptive mechanisms to acquire malignant characteristics necessary for cancer progression. Of these mechanisms, known as the “hallmarks of cancer”, protein homeostasis as regulated by the Endoplasmic Reticulum (ER) is a recognized process involved in cancer progression [4]. ER stress activates the Unfolded Protein Response (UPR) and has been implicated in a variety of cancers, including HCC. The UPR signal transduction cascade is directly activated as a response to prolonged ER stress conditions including nutrient deprivation, hypoxia, acidosis, drug-induced toxicity, and irradiation. In response to stressors, UPR response plays a major role in regulation of the expression of genes responsible for calcium and redox homeostasis, protein trafficking, ER quality control, autophagy, and lipid synthesis [5]. The UPR is inherently cell protective, aiming to alleviate damage and restore cellular homeostasis via transcriptional induction of specific molecular chaperones [6]. Due to the exposure of chronic stressors, cancer cells learn to adapt to prolonged ER stress by creating pro-survival alterations in the UPR signaling pathway and subsequently drive carcinogenesis [7]. Three major ER stress transducers, *IRE1, PERK,* and *ATF6*, are recognized as primary drivers of the UPR [5,8]. The role of protein homeostasis and the UPR in HCC has been studied to highlight of role of a specific UPR signal transducer, IRE1α, in HCC carcinogenesis via a metabolic inflammation mechanism [9]. A downstream regulator of the PERK-dependent branch of the UPR signaling pathway has been recognized to promote tumor cell proliferation via limiting oxidation DNA damage [10].

Hepatocellular carcinogenesis is etiologically linked to viral infection, chemical carcinogens, and other environmental and host factors that cause chronic liver injury. The accumulation of genomic alterations, DNA rearrangements, and chromosomal amplifications initiates the oncogenic progression of a normal hepatocyte to hepatocellular carcinoma [11]. In a related but distinct process from carcinogenesis, cancer progression is defined as the evolution of existing cancer from local regional advancement to metastatic disease, which is often recognized with clinical staging. Given the role of the UPR activation in carcinogenesis, we hypothesized that UPR activation could be recognized with pathological progression, mutational accumulation, clinical stage advancement, and survival in HCC.

Our previous work has reported the utility of scoring the genetic expression profile using gene set variation analysis (GSVA) to understand the relationship between signaling pathways and cancer biology in patients. For example, the G2M checkpoint pathway score identified margin-positive resection in pancreatic cancer [12] and metastasis in estrogen receptor-positive breast cancer patients [13], both resulting in poor survival. The DNA repair pathway was shown to be associated with cell proliferation and worse survival in HCC [14]. Given this background, we hypothesized in this study that the UPR was associated with unique characteristics and worse survival in HCC patients. To test our hypothesis, we analyzed a total of 655 HCC patients from The Cancer Genome Atlas (TCGA) Liver Hepatocellular Carcinoma (TCGA-LIHC; *n* = 358), GSE6764 (*n* = 75), GSE89377 (*n* = 107), and GSE76427 (*n* = 115) cohorts to examine the role of Unfolded Protein Response in clinical outcomes.

## 2. Materials and Methods

The hepatocellular carcinoma cohorts consisted of the mRNA-sequencing data of 358 hepatocellular carcinoma patients in The Cancer Genome Atlas (TCGA) Liver Hepatocellular Carcinoma cohort (TCGA_LIHC, *n* = 358), which was obtained from the Genomic Data Commons Data Portal (GDC). American Joint Committee on Cancer (AJCC) stage and pathological grade were obtained from GDC. We used the cohorts from Wurmbach et al. (GSE6764; *n* = 75) [15], Eun et al. (GSE89377; *n* = 107) [16], Brandon et al. (GSE56545; *n* = 42) [17], and Grinchuk et al. (GSE76427; *n* = 167) [18] to investigate the association between the DNA repair pathway scores and HCC patients’ clinicopathological characteristics and outcomes from the Gene Expression Omnibus (GEO) repository. Pathological classification of the samples in GSE6764 followed the guidelines of the International Working Party [19]. Four pathological HCC stages were defined: (i) Very early HCC (*n* = 8), which included well-differentiated tumors <2 cm in diameter with no vascular invasion/satellites (size range: 8–20 mm); (ii) early HCC (*n* = 10), which included tumors measuring <2 cm with microscopic vascular invasion/satellites; well- to moderately differentiated tumors measuring 2–5 cm without vascular invasion/satellites; or 2–3 well-differentiated nodules measuring <3 cm (size range: 3–45 mm); (iii) advanced HCC (*n* = 7), which included poorly differentiated tumors measuring >2 cm with microvascular invasion/satellites or tumors measuring >5 cm; and (iv) very advanced HCC (*n* = 10), which included tumors with macrovascular invasion or diffuse liver involvement. All genomic analyses used were log_2_ transformed normalized transcriptomic data. The average value was used for genes with multiple probes. Given that the TCGA and all GEO cohorts used in this study are de-identified in the public domain, approval from the Institutional Review Board was waived.

The UPR score was used as a surrogate for quantified UPR activity. The activity of UPR was quantified as the degree of enrichment of the “HALLMARK_UNFOLDED_PROTEIN_RESPONSE” gene set (Appendix A lists all the genes included in this gene set) defined and generated as one of the Hallmark gene set collections of the Molecular Signatures Database (MSigDB) [20] using the gene set variation analysis (GSVA) algorithm [21] in the Bioconductor package (version 3.10). We have previously reported the clinical relevance of multiple signaling pathways and responses using a similar approach [12,13,14,22,23,24,25,26,27,28,29,30,31,32,33,34,35,36,37,38,39].

The publicly available software (GSEA version 4.0.3) and the gene set enrichment analysis (GSEA) algorithm [40] was used in this study. Statistical significance was determined to a false discovery rate (FDR) of 0.25 as recommended by the GSEA software (Appendix A).

The xCell algorithm [41] was used to calculate the immune cell infiltration in the tumor microenvironment through transcriptomic data. The xCell data were obtained through the xCell website (https://xcell.ucsf.edu/, accessed on 23 February 2021), as we previously reported [22,23,24,25,26].

The score values of the intratumor heterogeneity, single-nucleotide variant (SNV) neoantigens, indel neoantigens, silent mutation, non-silent mutation, leukocyte fraction, lymphocyte infiltration, and interferon (IFN)- response score were calculated and published by Thorsson et al. [42]. Thorsson et al. [42] performed an extensive analysis of TCGA, which includes immune and genomic data from than 10,000 tumors across various cancer types. The study characterizes different immune subtypes by differences in immune cell signatures, extent of neoantigen load, overall cell proliferation, expression of immunomodulatory genes, and prognosis. The results and data analysis offers the structure of our methods.

The median value of the UPR score within cohorts was used to divide the patients into low and high UPR score groups. Statistical significance for comparison analysis between groups was determined to a *p*-value less than 0.05 by the Kruskal-Wallis test, the Mann-Whitney U test, and two-tail Fisher’s exact tests. Tukey-type boxplots showed the median and interquartile level values. R software (R Project for Statistical Computing, Appendix A) and Microsoft Excel (Appendix A) were used for all data analysis and data plotting.

## 3. Results

### 3.1. Unfolded Protein Response (UPR) Was Positively Correlated with Clinical Parameters of Carcinogenesis and Cancer Progression as Well as the AJCC Cancer Stage of HCC Patients

Unfolded Protein Response (UPR) was quantified by the GSVA score of the Molecular Signatures Database (MSigDB) Hallmark gene set using the methodology we previously described [12,13,14,27,28,29,30,31,32,33,34]. Based on previous basic research studies that have elucidated the role of UPR activation in carcinogenesis [9], we hypothesized that the UPR is enhanced through the step-wise progression of a normal liver into HCC in patients. To test this hypothesis, the UPR score was measured at each stage of histological progression—from normal liver, dysplasia, cirrhosis, low- and high-grade chronic hepatitis, to early and advanced HCC—in the GSE6764 and GSE89377 cohorts. UPR was significantly enhanced in early to advanced HCC compared to normal liver, dysplasia, cirrhosis, and very early HCC in the GSE6764 cohort (Figure 1A; *p* < 0.001). These results were replicated and validated in the GSE89377 cohort where the UPR was significantly enhanced in HCC compared with dysplasia, cirrhosis, and chronic hepatitis (Figure 1A; *p* < 0.001). As a measure of clinical cancer progression, UPR was also noted to be significantly enhanced in tumors with advanced American Joint Committee on Cancer (AJCC) staging (Figure 1B, *p* = 0.001).

### 3.2. UPR Was Positively Correlated with Multiple Parameters of Cell Proliferation, including Histological Grade, MKI67 Gene Expression and Enrichment of Cell Proliferation-Related Gene Sets by Gene Set Enrichment Assay (GSEA)

Given the finding that UPR was associated with HCC cancer progression, we decided to investigate the association with cancer cell proliferation. We found that UPR was positively correlated with a higher pathological grade as compared to a lower grade HCC tumor in the TCGA cohort (Figure 2A, *p* = 0.024). The expression of *MKI67*, a commonly used marker for cell proliferation in the clinical setting, was found to be significantly different between UPR groups when divided into low and high UPR score groups using the median value as a cut-off. The high UPR group was significantly associated with a high expression level of MKI67. In comparison, the low UPR group was associated with a low expression level of MKI67 (Figure 2C, *p* < 0.001).

The MSigDB Hallmark defines six gene sets as cell proliferation-related in GSEA. The UPR high HCC group significantly enriched five cell proliferation-related gene sets, including E2F targets, G2M checkpoint, MYC targets v1, MYC targets v2, and Mitotic spindle consistently in both TCGA and GSE76427 cohorts (Figure 2D, all False Detection Rate (FDR) < 0.25). Reviewed in their entirety, our results indicated that UPR was correlated with multiple measures of cell proliferation consistently, which validates the score as a parameter for cancer proliferation.

### 3.3. UPR Was Significantly Associated with Increased Mutational Load

As cancer cell proliferation is associated with high mutational rates [36], it was of interest to investigate the relationship between the UPR and mutation rate in HCC. Homologous recombination deficiency is representative of defective DNA repair and is a surrogate marker for increased mutational load. A high UPR score was significantly associated with homologous recombination deficiency, fraction altered, silent mutations, and non-silent mutations (Figure 3A, *p* = 0.002; Figure 3B, *p* < 0.001, *p* = 0.027, *p* = 0.030, respectively). The UPR score was not associated with single-nucleotide variants (SNV) neoantigens, or indel neoantigens (Figure 3B, *p* = 0.087, *p* = 0.290). This study found that a high UPR score was significantly associated with the mutation load.

### 3.4. There Was No Consistently Significant Association between the UPR and Immune Response or Immune Cell Infiltration

A high tumor mutational burden has been suggested to increase neoantigen production that can generate an anti-cancer immune cell infiltration in many types of cancers including HCC [36,43]. Thus, it was of interest to investigate the correlation between UPR and the immune response and immune cell infiltration, since a high mutation rate, but not neoantigens, was associated with high UPR. Although statistically significant (all FDR < 0.25), Normalized Enrichment Scores (NES) were uniformly low in all of the inflammation-related gene sets enriched to high UPR including TNFα signaling, IL6/STAT3 signaling, and complemented in both the TCGA and GSE76427 cohorts (Figure 4A). There was no significant increase in pro-cancer immune cell infiltration as estimated by the xCell algorithm in the UPR high HCC cohort, except for type2 Helper T cells in the TCGA cohort alone (Figure 4B). There was a noted significant increase in infiltration of type1 helper T cells, which are anti-cancer cells, in the TCGA cohort alone (Figure 4B, *p* = 0.017). Low UPR score had a noted increase in M2 macrophages, a pro-cancer immune cell, in the TCGA cohort alone (Figure 4B, *p* = 0.029). Interestingly, with a high degree of mutational variation in the analyzed cohorts and a presumed increase of antigen presentation, there was no consistent immune cell infiltration into the tumor microenvironment of UPR high HCC. There was no correlation between the UPR score and expression of PD-1 or PD-L1 (Appendix A).

### 3.5. UPR Was Not Consistently Associated with Angiogeneis across Independent Cohorts of HCC

It was of interest to investigate the association between UPR and angiogenesis because UPR has been shown to mediate angiogenesis through the regulation of transcription factors [44]. The angiogenesis gene set was not enriched to HCC with high UPR in either of the TCGA and GSE76427 cohorts (Figure 5, NES = 1.16 and FDR = 0.268; NES = 1.15 and FDR = 0.306). HCC with high UPR was also associated with a decreased infiltration of lymphatic vessel related cells, including endothelial cells and lymphatic endothelial cells, as well as adipocytes in the TCGA cohort (Figure 5, *p* < 0.001, *p* = 0.016, *p* < 0.001), but this result was not validated in the GSE76427 cohort.

### 3.6. The Unfolded Protein Response Score Was Consistently Associated with Worse Survival across Independent Cohorts of HCC

As previous studies have reported that enhanced UPR was associated with worse survival in other cancer, most notably in glioblastoma multiform [45,46], it was of interest to investigate a similar relationship with survival in HCC patients. We analyzed the UPR score as related to overall survival (OS), disease-free survival (DFS), risk-free survival (RFS), and disease-specific survival (DSS) in the TCGA, as well as RFS in GSE76427 cohorts. Each cohort was divided into low and high UPS groups using the median value. The high UPS was significantly associated with worse OS, DFS, RFS, and DSS in all cohorts of TCGA, and poor RFS in GSE62452 (Figure 6). In addition, through univariate and multivariate cox regression analysis using OS in the TCGA cohort, the UPR score and AJCC stage were demonstrated to be independent prognostic factors for HCC patients (Appendix A). These results suggest that Unfolded Protein score is able to quantify the biological aggressiveness of HCC and has the potential to be used as a prognostic biomarker for survival in HCC. We have previously published that cell proliferation-related scores that are associated with patient survival including MYC [30], G2M checkpoint [12], and E2F target [32] scores in breast cancer.

We also reported that the G2M checkpoint pathway alone is associated with drug response and survival among cell proliferation-related pathways in pancreatic cancer [12]. However, we have never compared or analyzed which of these scores correlate most with survival in HCC. To this end, we analyzed the clinical benefit of UPR and each cell proliferation-related gene sets, including E2F targets, G2M checkpoints, MYC target v1 and v2, MITOTIC spindle, and MKI67 expression by a Cox Proportional Hazards model. As demonstrated in Appendix A, the hazard ratio of UPR was the highest among all the biomarkers, which offers it a higher degree of correlation over our previously analyzed cell proliferation-related scores.

## 4. Discussion

In this study, we looked at the association of the unfolded protein response (UPR), as measured by the UPR Score, with clinical relevance in HCC. Our findings suggested that the UPR was positively correlated with each histological progression in the carcinogenesis and progression of HCC. The UPR high HCC cohort was significantly associated with multiple parameters of cell proliferation including histological grade, *MKI67* gene expression, and enrichment of cell proliferation-related gene sets by GSEA and mutational load. On the contrary, the UPR high HCC cohort was not associated with angiogenesis or increased immune activity. The UPR was also consistently associated with worse survival across independent cohorts of HCC. To our knowledge, this is the first study to investigate the clinical relevance of the unfolded protein response in HCC.

The UPR signaling network operates as a pro-oncogenic mechanism that drives several aspects of cancer progression by increasing cancer cell survival and adapting to intrinsic changes and environmental challenges. Most evidence suggests that the UPR is involved in most hallmarks of cancer, including cell proliferation, immune evasion, angiogenesis, and treatment resistance [4,5]. Chronic environmental stressors, including nutrient deprivation, hypoxia, acidosis, drug-induced toxicity, and irradiation, drives the UPR to adopt a pro-survival mechanism that is co-opted by cancer cells for continued proliferation and survival. These changes are demonstrated in gene expression alteration and changes in protein signal transduction. The role of protein homeostasis and the UPR in HCC has been studied to highlight of role of a specific UPR signal transducer, IRE1α, in HCC carcinogenesis via a metabolic inflammation mechanism [9]. A downstream regulator of the PERK-dependent branch of the UPR signaling pathway has been recognized to promote tumor cell proliferation via limiting oxidation DNA damage [10]. Our findings highlighted a significant correlation between UPR activation through each histological stage in the carcinogenesis of normal liver tissue to hepatocellular cancer. These results offer a clinical observation of the previously defined pre-clinical role of the UPR in carcinogenesis. It is tempting to speculate that this observation offers a translation target for targeted therapy to interrupt hepatocellular carcinogenesis.

Cancer progression is defined as the advancement of existing cancer, from local regional advancement to metastatic disease, which is measured by parameters of clinical staging. Utilizing an in vivo preclinical model, one study noted that cancer failed to progress if a downstream UPR-activated chaperone protein was suppressed, directly demonstrating the role of UPR-related proteins in cancer progression [47]. With the observed preclinical data that elucidates the role of UPR in cancer progression, it can be reasonably assumed that the UPR would be associated with cancer stage. Our study observed the significant enrichment of the UPR in advancing AJCC staging, with high UPR enrichment in stage IV HCC and lower UPR enrichment in stage I HCC.

The UPR is difficult to quantify in the clinical setting. Prior studies have analyzed the actions of three major ER stress transducers, *IRE1, PERK,* and *ATF6*, as a measure of the UPR and its adaptive response to ensure cell survival [5,8]. Without gene expression profiling and protein transduction studies, there is a lack of a translational method to quantify the UPR. GSEA allows us to capture genetic activation by the UPR across the transcriptomes of multiple large HCC patient cohorts and create a quantifiable entity in the UPR score. We can then use this score to study its association with common clinical parameters of cancer proliferation and progression. The Hallmark gene sets utilized in this study are widely accepted and recognized as representing well-defined biological processes. The aim of this study was to investigate the clinical relevance of the existing Hallmark Unfolded Protein Response gene set, and not to generate a novel gene set that reflects the UPR. We have previously reported the association of several pathways with clinical outcomes in various cancers [12,13,31,32].

The common histological assessments of grade and *MKI67* expression are used as clinical correlates to recognize degrees of cell proliferation in cancer. A prior study of UPR activation demonstrated that downstream UPR signal transducers were overexpressed more frequently in the higher-grade breast cancers than in lower-grade breast cancers indicating that activation of the UPR can correlate with a clinically more aggressive phenotype [48]. This study observed a similar correlation in that UPR gene set activation was highly enriched with higher grade HCC, and not as highly enriched with lower grade HCC. In addition, the enrichment of cell proliferation gene sets supports the finding of significant UPR association with increased cell proliferation as measured by mKI67 expression.

Neoplastic progression requires several genetic alterations and mutations that allow the cell to ignore growth controls and disable apoptotic signaling. The accumulation of a mutational load is associated with a high rate of proliferation as we have previously observed in breast cancer [36], and so it was interest to investigate if a high mutational load cancer is associated with a high UPR. This study found that a high UPR score was significantly associated with the mutation load. In general, the tumor microenvironment (TME) is manipulated by cancer cells by the release of pro-inflammatory mediators. This results in the recruitment of immune cells, which are involved in angiogenesis, invasion, as well as metastasis [49]. It has been shown that this change in the TME is a form of ER stress that is influenced by sustained activity of IRE1α, a main sensor of the UPR signaling pathway [50]. In addition, in vitro experimental data have shown that macrophages activate UPR when cultured with ER-stressed cancer cells. Furthermore, these macrophages recapitulate, amplify, and expand the proinflammatory response of cancer cells [50]. Thus, we hypothesized that UPR activation would be associated with inflammation in the TME. Interestingly, we did not observe a consistent association between UPR activation and an immune response. In addition, as higher mutational load cancer has been associated with increased neoantigen presentation and subsequent immune cell infiltration [36], this same relationship was not observed in HCC cohorts.

Angiogenesis is crucial to the progression of cancer [31,37,38,39,51,52,53]. Rapid tumor growth and inadequate vascularization creates environmental stress that propagates the activation of stress response pathways, include the unfolded protein response. Studies have indicated that cells suffering from insufficient blood supplies experience ER stress and activate the UPR to help cancer cells continue to grow and spread rapidly [54,55]. The UPR has also been shown to play a role in mediating angiogenesis through the regulation of VEGFA transcription factor [44]. Though much pre-clinical evidence points to a correlative relationship, it is conceivable that the activation of the angiogenic and UPR pathways could synergize in some cases and be antagonistic in others [56]. Interestingly, our analysis did not show a consistent relationship between the UPR and angiogenesis across TCGA and GSE76427 cohorts.

This study observed the association of the UPR with clinical parameters of cancer aggressiveness and progression. By extension, it was reasonable to assume that high UPR activation would also be associated with worse survival. Our results indicated a statistically significant and validated correlation with recurrence-free survival across two cohorts of HCC patients. Overall survival was also worse with high UPR activation in the TCGA cohort. Though the aim of the study was not to report a new mechanism of the unfolded protein response in HCC, it does highlight the ability of the UPR score to predict survival in HCC.

The current study has obvious limitations. The uses of databases have an inherent inability to analyze tissue samples directly, thus limiting our ability to perform further comprehensive histological analysis. As we used publicly available cohorts, our results may not reflect the heterogeneity among the patients due to the limited sample size of the databases. Although we believe that the statistical significance is real when the difference exists despite the small sample size, we may not be inclusive of all findings. Another limitation of this study is that is does not accurately reflect the composition of the tumor microenvironment as represented by immunological milieu of the original cancer tissue, which cannot be reliably replicated outside of in vivo experiments. Finally, the cohorts we access do not contain treatment data and it is assumed that the patients underwent standard-of-care treatments. In the future, our result needs validation with prospectively analyzing UPR score in patient samples with treatment data.

## 5. Conclusions

The unfolded protein response (UPR) was associated with carcinogenesis and progression of HCC, with multiple parameters of cell proliferation, including histological grade, MKI67 gene expression, and cell proliferation-related gene sets, with increased mutational load, but not with immune infiltration nor angiogenesis, and with worse survival across independent cohorts of HCC. Thus, the UPR score may be useful as a biomarker to predict prognosis and to better understand HCC.

## Figures and Tables

**Figure 1 cancers-13-04443-f001:**
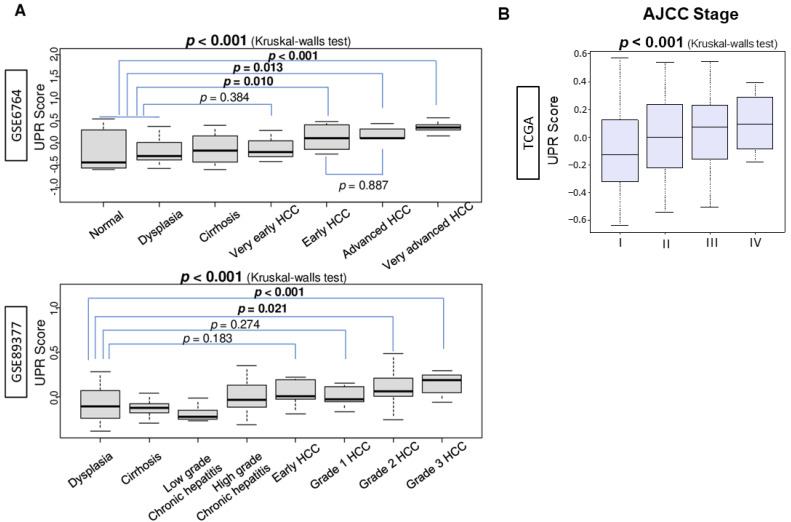
Association between Unfolded Protein Response (UPR) and hepatocarcinogenesis and progression. (**A**) Boxplots of the comparison of the unfolded protein response score by multistep hepatocarcinogenesis, including normal liver tissue (*n* = 8), dysplasia (*n* = 17), cirrhosis (*n* = 13, very early hepatocellular carcinoma (HCC) (*n* = 8), early HCC (*n* = 10), advanced HCC (*n* = 7), and very advanced HCC (*n* = 10) defined by the GSE764 cohort (*n* = 75); and dysplasia (*n* = 35), cirrhosis (*n* = 12), low-grade (*n* = 8) and high-grade (*n* = 12) chronic hepatitis, early HCC (*n* = 5), and grades 1–3 (*n* = 9, 12, and 14, respectively) of HCC defined by the GSE98377 cohort (*n* = 107). The *p*-value of normal/dysplasia vs. each class of HCC analyzed using the Mann–Whitney U test in the GSE6764 cohort. The *p*-value of dysplasia vs. each class of HCC analyzed using the Mann–Whitney U test in the in GSE89377 cohort. The overall *p*-value was calculated using a Kruskal–Wallis test. (**B**) Boxplot of the comparison of the unfolded protein response and the American Joint Committee on Cancer (AJCC) stage I-III (*n* = 166, 81, and 84, respectively) in the TCGA cohort. The overall *p*-value was calculated using a Kruskal–Wallis test.

**Figure 2 cancers-13-04443-f002:**
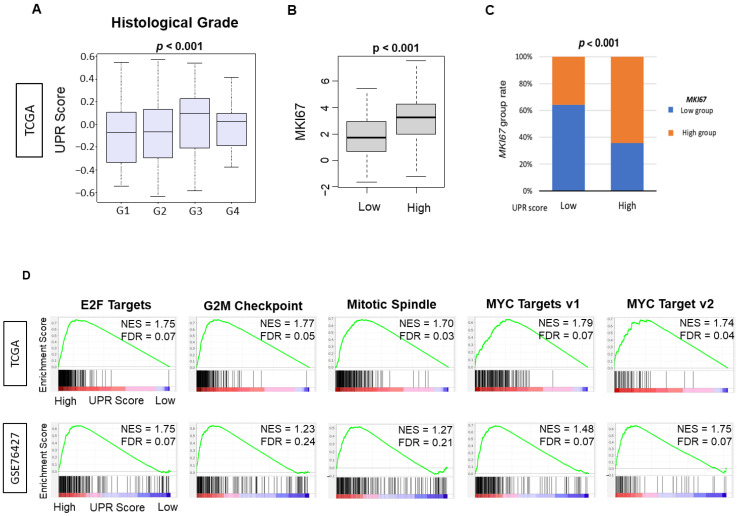
Association between Unfolded Protein Response and histological grade, MKI67 gene expression and cell proliferation-related gene sets. (**A**) Boxplot of the comparison of the unfolded protein response score with histological grade G1- 4 (*n* = 53, 168, 121 and 11, respectively). The *p*-value was calculated using a Kruskal–Wallis test. (**B**) Boxplot of the comparison of the high vs. low unfolded protein response and the MKI67 gene expression (both *n* = 179) in the TCGA cohort. The *p*-value was calculated using the Mann–Whitney U test. (**C**) Bar plot of the comparison of low and high groups of the unfolded protein response score and MKI67 expression. The *p*-value was calculated using Fisher’s exact test. (**D**) Gene set enrichment analysis (GSEA) of the Hallmark gene sets by high vs. low unfolded protein response score of HCC in the GSE76427 and TCGA cohorts. Enrichment plots with the normalized enrichment score (NES) and false discovery rate (FDR) for proliferation-related gene sets. An FDR of 0.25 was used to determine statistical significance as recommended by the GSEA software (version 4.1.0, accessed on 27 February 2021) (Appendix A).

**Figure 3 cancers-13-04443-f003:**
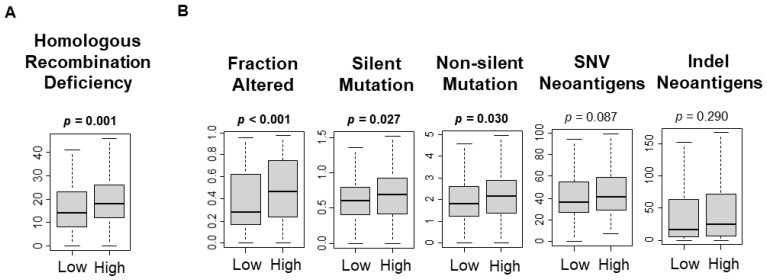
Association between Unfolded Protein Response and mutation-related scores. Boxplots of the comparison of the high vs. low unfolded protein response and (**A**) homologous recombinant defects score; (**B**) fraction altered, silent mutation, non-silent mutation, single-nucleotide variants (SNV) neoantigens, and indel neoantigens. The *p*-value was calculated using the Mann–Whitney U test. Bold format: significant *p* values.

**Figure 4 cancers-13-04443-f004:**
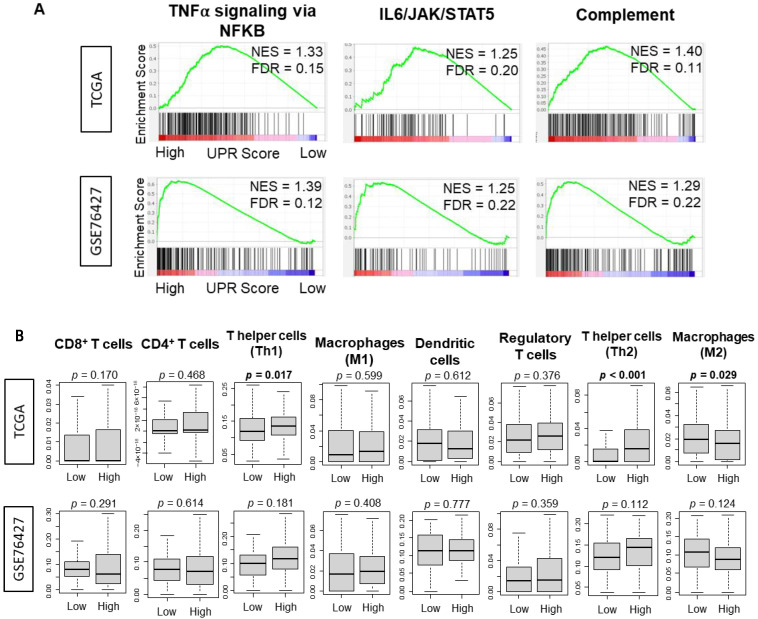
Association between Unfolded Protein Response and inflammation-related gene sets as well as infiltrating immune cells in the TCGA and GSE76427 cohorts. (**A**) Enrichment plots with the normalized enrichment score (NES) and false discovery rate (FDR) for the TNFα signaling via NFKB, IL6/JAK/STAT5, and complement gene sets of the Hallmark gene sets. (**B**) Boxplots of the anti-cancer immune cells including CD8^+^ T cells, CD4^+^ T cells, T helper type 1 (Th1) cells, M1 macrophages and dendritic cells; and pro-cancer immune cells including regulatory T cells, T helper type (Th2) cells, and M2 macrophages. The *p*-value was calculated using the Mann-Whitney U test. Bold format: Significant *p* values.

**Figure 5 cancers-13-04443-f005:**
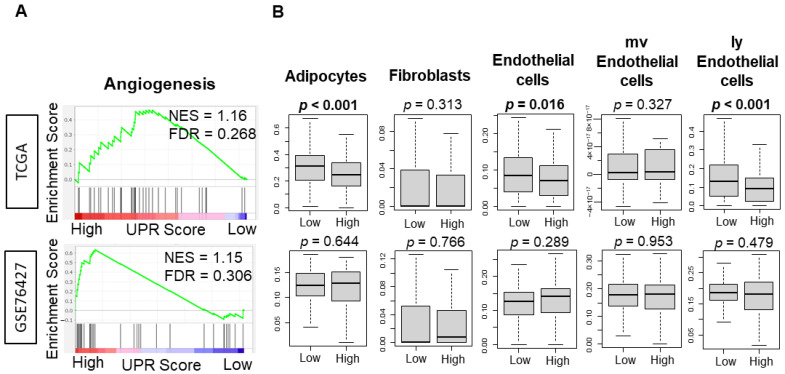
Association between Unfolded Protein Response and angiogenesis in the TCGA and GSE76427 cohorts. (**A**) Enrichment plots with the normalized enrichment score (NES) and false discovery rate (FDR) for the angiogenesis gene sets of the Hallmark gene sets. (**B**) Boxplots of the angiogenesis related gene sets adipocytes, fibroblasts, endothelial cells, mother vessel (mv) endothelial cells, and lymphatic (ly) endothelial cells. The *p*-value was calculated using the Mann-Whitney U test. Bold format: Significant *p* values.

**Figure 6 cancers-13-04443-f006:**
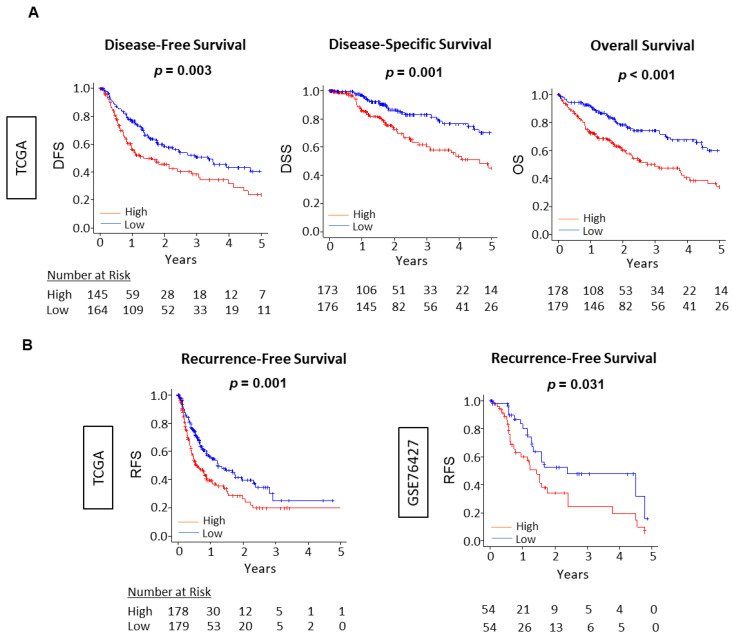
Association between the Unfolded Protein Response and HCC patient survival. (**A**) Kaplan–Meier survival curves comparing the low and high unfolded protein response score in HCC to demonstrate disease-free survival, disease-specific survival, and overall survival in the TCGA cohort (*n* = 358). (**B**) Kaplan–Meier survival curves comparing the low and high unfolded protein response score in HCC to demonstrate recurrence-free survival in the TCGA and GSE76427 (*n* = 167) cohorts. We divided the cohort into low and high unfolded protein response score groups using the median value as the cut-off. The *p*-value was calculated using a log rank test.

## Data Availability

All data was obtained from deidentified publicly available databases, TCGA cohort from the Genomic Data Commons Data Portal (GDC) (https://portal.gdc.cancer.gov/, accessed on 27 August 2021), and GSE6764, GSE89377, GSE56545, and GSE76427 from the Gene Expression Omnibus (GEO) repository (https://www.ncbi.nlm.nih.gov/geo/, accessed on 27 August 2021).

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
