# Peer review of "The Unfolded Protein Response Is Associated with Cancer Proliferation and Worse Survival in Hepatocellular Carcinoma"

_cancers, 2021, doi:10.3390/cancers13174443_

Round 1

Reviewer 1 Report

1. This is a clear and well-written manuscript. The introduction is relevant, however, it would be nice to expand a bit more on the UPR activation and its transducers. It would be also interesting to discuss more the role of ER stress in HCC and how the signaling pathways are altered due to the exposure of chronic stressors. 

2. The methods are well described and the results compelling.

3. The three main limitations of this study have been acknowledged  by the authors: it would be very interesting to confirm the above findings within tissue samples of patients or at least to have a larger cohort/database and to include treatment data. However, it is still a well-designed study that highlights the importance of UPR score as a prognostic biomarker in HCC.

Reviewer 2 Report

In this manuscript, the authors demonstrated that a high UPR score might associate with different important clinical features of HCC. And UPR score might be further developed to be a useful biomarker for HCC patients. Before publication, they are some important issues the authors should clarify and address

  1. Since UPR score is essential in this manuscript, the authors should clearly describe how they calculated UPR score for each cohort and if UPR scores in the four cohorts are comparable. In addition, the authors need to mention how they define high and low UPR scores.
  2. In line 150, the authors mentioned that UPR scores increased from early HCC to advanced HCC compared with normal and dysplasia according to Kruskal Walls test (KW test). KW test (P < 0.05) can only there is/are group(s) with mean value(s) different significantly but the test cannot tell which of the particular groups show a significant difference. The authors need to employ another statistical test to determine the difference between groups (normal/dysplasis vs early HCC; normal/dysplasis vs advanced HCC; early HCC vs advanced HCC).  
  3. In line 175, the results regrading to Figure 2B, I suppose the authors employed Mann-Whitney U test. If yes, the test can only tell mKi67 scores between low UPR and high UPR groups are significantly different. However, it cannot conclude if there is any association between mKi67 score and UPR score. For the association, the authors need to employ another statistical analysis such as the chi-square test which can determine positive and negative association.
  4. In figure 3, the authors did not mention the statistical test they used. Also, the authors did not mention the relationship between mutational rate and homologous recombination deficiency. It is hard for the reader to understand the logic and the importance of the results from figure 3.
  5. Again, the authors did not mention the statistical test used in figure 4b, figure 5.
  6. In figure 6, the authors demonstrated HCC patients with high UPR scores would have poorer survival outcomes. I suggest the authors perform univariate and multivariate cox regression analysis to determine if UPR score would be an independent prognostic factor for general HCC patients or HCC patients with particular clinical features.  
  7. Finally, the author might need to discuss the advantages of using UPR score as a biomarker over other currently used markers. If UPR score cannot give extra and important information, no one will use it.

Reviewer 3 Report

The authors show that the UPR protein response is a prognostic marker for hepatocellular carcinoma. They used the TCGA HCC database and other Geo resources to correlate UPR with overall or disease-free HCC disease survival. The work applies various bioinformatics tools to existing datasets, thus a good value for biomarker discovery and future mechanistic studies.  Some of the comments below.

How does UPR score differ or become more beneficial from myc or cell proliferation gene signature should be discussed. Does the UPR score predict outcome better than, say ki67 score? The unique/special nature of the UPR score is not immediately evident.  Can a ROC curve be included?

How does UPR score correlate with PD-L1/PD1 expression?

Being a heavy bioinformatics paper, the authors can provide details of the primary literature. In addition, a table of software with references and version and github pages can be included.

The UPR score should be discussed in a bit of detail. Even though they refer to their previous work, it should be discussed since this is the crux of the paper. Similar details are needed for HRD scores and cell-type identification, for example, in Figure 5.

Round 2

Reviewer 2 Report

The authors have already addressed all my concerns.  Also, the missing information such as the details of the statistical tests used has been included in the revised manuscript. The authors improve the manuscript significantly.